# Multi-Parametric Radiomic Model to Predict 1p/19q Co-Deletion in Patients with IDH-1 Mutant Glioma: Added Value to the T2-FLAIR Mismatch Sign

**DOI:** 10.3390/cancers15041037

**Published:** 2023-02-07

**Authors:** Shingo Kihira, Ahrya Derakhshani, Michael Leung, Keon Mahmoudi, Adam Bauer, Haoyue Zhang, Jennifer Polson, Corey Arnold, Nadejda M. Tsankova, Adilia Hormigo, Banafsheh Salehi, Nancy Pham, Benjamin M. Ellingson, Timothy F. Cloughesy, Kambiz Nael

**Affiliations:** 1Department of Radiological Sciences, David Geffen School of Medicine at University of California Los Angeles, 757 Westwood Plaza, Suite 1621, Los Angeles, CA 90095, USA; 2Department of Radiology, Kaiser Permanente Fontana Medical Center, Fontana, CA 92335, USA; 3Department of Bioengineering, University of California Los Angeles, Los Angeles, CA 90095, USA; 4Department of Pathology and Laboratory Medicine, Icahn School of Medicine at Mount Sinai, New York, NY 10029, USA; 5Department of Neurology, Icahn School of Medicine at Mount Sinai, New York, NY 10029, USA; 6UCLA Brain Tumor Imaging Laboratory, Department of Radiological Sciences, David Geffen School of Medicine at University of California Los Angeles, Los Angeles, CA 90095, USA; 7Department of Neurology, David Geffen School of Medicine at University of California Los Angeles, Los Angeles, CA 90095, USA

**Keywords:** deep learning, diffuse glioma, radiogenomic, T2-FLAIR mismatch, 1p/19q co-deletion

## Abstract

**Simple Summary:**

The T2-FLAIR mismatch sign has shown promise in determining IDH mutant 1p/19q non-co-deleted diffuse gliomas with a high specificity and modest sensitivity. We aim to develop a multi-parametric radiomic model using MRI to predict the 1p/19q co-deletion status in patients with newly diagnosed IDH1 mutant diffuse glioma. In this retrospective study, patients with a diagnosis of IDH1 mutant gliomas with a known 1p/19q status who had preoperative MRI were included. T2-FLAIR mismatch was evaluated independently by two board-certified neuroradiologists. eXtremeGradient Boosting (XGboost) classifiers were used for model development. A total of 103 patients included for model development and 18 patients for external testing validation. The diagnostic performance (sensitivity/specificity/accuracy) in determination of 1p/19q co-deletion status were 59%/83%/67% (training) and 62.5%/70.0%/66.3% (testing) for T2-FLAIR mismatch sign. This was significantly improved (*p* = 0.04) using the radiomics model to 77.9%/82.8%/80.3% (training) and 87.5%/89.9%/88.8% (testing), respectively. The proposed radiomic model provides much needed sensitivity to the highly specific T2-FLAIR mismatch sign in determination of 1p/19q non-co-deletion status and improves overall diagnostic performance of neuroradiologists when used as an assistive tool.

**Abstract:**

Purpose: The T2-FLAIR mismatch sign has shown promise in determining IDH mutant 1p/19q non-co-deleted gliomas with a high specificity and modest sensitivity. To develop a multi-parametric radiomic model using MRI to predict 1p/19q co-deletion status in patients with newly diagnosed IDH1 mutant glioma and to perform a comparative analysis to T2-FLAIR mismatch sign+. Methods: In this retrospective study, patients with diagnosis of IDH1 mutant gliomas with known 1p/19q status who had preoperative MRI were included. T2-FLAIR mismatch was evaluated independently by two board-certified neuroradiologists. Texture features were extracted from glioma segmentation of FLAIR images. eXtremeGradient Boosting (XGboost) classifiers were used for model development. Leave-one-out-cross-validation (LOOCV) and external validation performances were reported for both the training and external validation sets. Results: A total of 103 patients were included for model development and 18 patients for external testing validation. The diagnostic performance (sensitivity/specificity/accuracy) in the determination of the 1p/19q co-deletion status was 59%/83%/67% (training) and 62.5%/70.0%/66.3% (testing) for the T2-FLAIR mismatch sign. This was significantly improved (*p* = 0.04) using the radiomics model to 77.9%/82.8%/80.3% (training) and 87.5%/89.9%/88.8% (testing), respectively. The addition of radiomics as a computer-assisted tool resulted in significant (*p* = 0.02) improvement in the performance of the neuroradiologist with 13 additional corrected cases in comparison to just using the T2-FLAIR mismatch sign. Conclusion: The proposed radiomic model provides much needed sensitivity to the highly specific T2-FLAIR mismatch sign in the determination of the 1p/19q non-co-deletion status and improves the overall diagnostic performance of neuroradiologists when used as an assistive tool.

## 1. Introduction

Gliomas are infiltrative brain neoplasms that are classified by *isocitrate dehydrogenase* (*IDH*) gene status and the whole-arm co-deletion of chromosome arms 1p and 19q [1]. In the new 2021 WHO guidelines, IDH1 mutation is a histopathological characterization of all lower-grade gliomas and, of those, 37–50% carry the 1p/19q co-deletion [2,3,4]. Patients with 1p/19q co-deleted tumors have a better treatment response to radiation and chemotherapy and a longer progression-free and overall survival [5,6].

Therefore, the more accurate and early detection of 1p/19q co-deletion status has prognostic and treatment implications.

Over the last decade, imaging markers on MRI have been explored as non-invasive signatures that could potentially serve as surrogates for some of the molecular alterations including 1p/19q co-deletion status in patients with glioma. This in turn may change the preoperative differential diagnosis of these tumors and potentially aid treatment decision processes [7]. In recent years, the “T2-FLAIR mismatch” has been investigated as a noninvasive imaging marker for 1p/19q co-deletion status [8,9] that is easily discernible and broadly available on standard routine MRI sequences. This sign is defined by a completely or almost completely homogeneous hyperintense signal on the T2 sequence and simultaneously completely or almost completely homogeneous hypointense signal on the FLAIR sequence with a complete or near-complete hyperintense peripheral rim on FLAIR.

Patel et al. [8] first described the T2-FLAIR mismatch sign as a positive predictor for 1p/19q non-co-deletion status within IDH1 mutant gliomas with PPV of up to 100%. However, the clinical utility of this mismatch sign is limited by its low sensitivity of about 55% and modest inter-observer agreement [8,10,11]. Furthermore, the sign was only seen in 15 out of 125 cases of LGGs with the 1p/19q non-co-deletion. In a recent study by Foltyn et al. [12], the T2-FLAIR mismatch sign showed a 100% specificity and PPV but only 10.9% sensitivity for the detection of 1p/19q co-deletion status. The presence of calcification has also emerged as a promising imaging predictor for 1p/19q co-deletion status as seen in oligodendroglioma with a specificity of up to 97%; however, again, sensitivity remains low around 40% [13,14].

Due to the low sensitivity of identifiable radiological features on conventional imaging such as T2-FLAIR mismatch or calcification, recent interest has been placed on exploring radiomic prediction. Zhou et al. [15] first described radiomic prediction of the co-deletion status, demonstrating AUC of up to 0.88. Several subsequent studies have also shown positive results for radiomics for the prediction of 1p/19q co-deletion status with or without the inclusion of clinical models, reporting AUCs ranging from 0.87 to 0.89 [16,17,18,19,20,21]. The diagnostic performance of radiomics in the context of the routinely used T2-FLAIR mismatch sign in the same patient cohort has however not been systematically reported. The ability to accurately predict 1p/19q co-deletion status noninvasively can guide clinical management and assess prognosis. In addition, histopathological assessment of 1p/19q can occasionally be erroneous using Fluorescence In Situ Hybridization (FISH) [22]. Therefore, there may be a role for neuroimaging prediction to address these occasional discrepancies related to FISH error.

In this study, we aim to construct a radiomic model from preoperative MRI and evaluate its diagnostic performance compared to the routinely used T2-FLAIR mismatch sign by: (1) developing a radiomic model using eXtremeGradient Boosting (XGboost) classifiers from preoperative FLAIR images to predict the 1p/19q co-deletion status in patients with IDH mutant diffuse glioma; (2) performing a comparative analysis against the T2-FLAIR mismatch sign in predicting 1p/19q co-deletion status both in training and in an external validation cohort; and (3) through using the proposed radiomic model as a computer-assisted tool, assessing its added value to the diagnostic performance of the T2-FLAIR mismatch sign from a neuroradiologist.

## 2. Materials and Methods

### 2.1. Patient Selection

This retrospective study was approved by an institutional review board and informed consent was waived. Patients with initial diagnosis of gliomas between January 2016 to September 2018 were reviewed. Patients were included if they (1) had diagnosis of IDH1 mutant diffuse gliomas with known 1p/19q co-deletion status and (2) had preoperative MRI including T2WI and FLAIR within 30 days from the biopsy (Figure 1). 120 patients were excluded for not having preoperative MR within 30 days and an additional 20 patients were excluded for not having T2WI or FLAIR sequences. An external validation cohort was also included from an outside institution under an approved institutional review board. This cohort has identical inclusion and exclusion criteria.

### 2.2. Histopathologic Data

Tissue samples were obtained from patients undergoing targeted tissue biopsy, as part of routine clinical care and diagnostic neuropathology and molecular evaluation. Immunohistochemistry (IHC) was used to detect mutant status of *IDH1* (specifically IDH1^R132H^ immunoreactivity). Next-generation sequencing (NGS) was performed to confirm mutational status and in rare cases identified alternative IDH mutations which were not picked up by IHC. For determination of 1p/19q co-deletion status, dual-color interphase FISH hybridization was performed on paraffin-embedded tissue using locus-specific identifier probe sets 1p36/1q25 and 19q13/19p13, scored in 200 cells, for which a ratio of 1p/1q and 19q/19p, both less than 0.75, was considered as “co-deleted”/positive for co-deletion.

### 2.3. Image Analysis

Image acquisition was performed using a standardized preoperative brain tumor MRI protocol in accordance with consensus recommendations for a standardized brain tumor imaging in clinical trials [23].

Two board-certified neuroradiologists with 4 and 5 years of post-fellowship experience blinded to the results of 1p/19q status reviewed the images independently for T2-FLAIR mismatch sign. T2-FLAIR mismatch sign was determined based on strict inclusion criteria used in prior studies [8,9] as follows: (1) on T2WI, there was complete or near-complete and homogeneous hyperintense signal, and (2) on FLAIR, there was hypointense signal except for a thin, hyperintense peripheral rim. The discrepancies between observers were resolved during a consensus read session. The consensus reads were used to assess the diagnostic performance of T2-FLAIR mismatch sign.

For external validation, the T2-FLAIR mismatch were evaluated by both readers jointly and in the same session. The observers were blinded to the 1p/19q status.

In a separate experiment, Neuroradiologist 1 was asked to rescore the T2-FLAIR mismatch assignments while having access to the initial scores and radiomics-predicted scores. This experiment was performed several months after the initial readout to reduce recall bias. The observer remained blinded to the histopathology results.

### 2.4. Texture Analysis

Tumor segmentation was performed manually on FLAIR images using volume-of-interest (VOI) analysis on commercially available FDA-approved software (Olea Sphere, SP23, Olea Medical SAS, La Ciotat, France) with automated, standardized preprocessing and normalization through the software for each case. Subsequently, a VOI was generated using a voxel-based signal-intensity-threshold method subsuming the entire region of FLAIR hyperintensity. Using Olea Sphere texture analysis plug-in, a total of 92 texture features were extracted from each VOI in every patient and used for statistical analysis and radiomic model development. Texture features included nineteen first-order metrics, such as the mean, standard deviation, skewness, and kurtosis. Second-order metrics included twenty-three gray-level run-length matrices, sixteen gray-level run-length matrices, fifteen gray-level size-zone matrices, five neighboring gray tone difference matrices, and fourteen gray-level dependence matrices [24,25,26].

### 2.5. Statistical Analysis

Statistical analysis was performed using R software (version 4.1.3; htttps://www.r-project.org/ (accessed on 23 September 2022)). The interobserver agreement for T2-FLAIR mismatch sign was assessed using a weighted kappa test. The diagnostic performance of neuroradiologists using T2-FLAIR mismatch sign was evaluated in prediction of 1p/19q status using receiver-operating characteristic (ROC) curves and area under the curve (AUC) analysis for training and testing data sets.

From texture features, an imaging model was developed to predict 1p/19q co-deletion status using eXtremeGradient Boosting (XGboost) [27]. First, five-repeated 10-fold cross-validation and grid search was used to pick the best hyperparameters set for XGboost model and the top 6 most important texture features based on Gini index. Second, the model was trained using the fine-tuned hyperparameters and the 6 features. Internal leave-one-out-cross-validation (LOOCV) and external validation performance results were reported. Optimal thresholds were determined to maximize sensitivity and specificity for each biomarker using the top-left-corner cut-off point of the ROC curve. The criterion is defined as
$$\mathrm{cutoff}=\;\min\{{\left(1-\mathrm{sensitivity}\right)}^{2}+{\left(1-\mathrm{specificity}\right)}^{2}\}$$

The aforementioned experiments were replicated 100 times using different random seed to account for randomness. Performance ranges were reported in a mean +/− standard deviation form.

The accuracy of the final radiomic model was evaluated using ROC and AUC analyses for training and testing data sets. The ROC comparison between radiomics and neuroradiology interpretation was performed using a DeLong test.

## 3. Results

### 3.1. Clinical Characteristics of Patient Population

From our internal data set that was used for model development, a total of 103 patients, 66 male and 37 female, included (Table 1). The average patient age was 41.1 ± 13.2 (mean ± SD). The tumor location was identified in the frontal lobe (*n* = 57), parietal lobe (*n* = 18), temporal (*n* = 23), occipital (*n* = 0), and cerebellum (*n* = 5). The status of 1p/19q was non-co-deleted in 68 patients and co-deleted in 35 patients.

External validation was performed in a cohort of 18 patients from an outside institution, 10 males and 8 females. Average patient age was 43.2 ± 13.0 (mean ± SD). The status of 1p/19q co-deletion was non-co-deleted in 10 patients and co-deleted in 8 patients.

### 3.2. T2-FLAIR-Mismatch-Based Prediction of 1p/19q Co-Deletion Status

Among 103 patients, the number of patients with T2-FLAIR mismatch/match was 43/60 for Neuroradiologist 1 (4 years of experience) and 62/41 for Neuroradiologist 2 (5 years of experience). The interobserver agreement for defining T2-FLAIR mismatch was modest (k = 0.49, 95% CI: 0.33–0.64). Following consensus reads, a total of 58 T2-FLAIR mismatches were identified while 45 patients were identified as having a T2-FLAIR match. This resulted in a diagnostic performance (AUC/sensitivity/specificity/accuracy) of 0.70/57%/83%/67%.

Examples of true positive, true negative, false positive, and false negative results for T2-FLAIR mismatch using consensus reads against 1p/19q co-deletion status are shown in Figure 2.

### 3.3. Radiomic-Based Prediction of 1p/19q Co-Deletion Status

Using a radiomic-generated model from a combination of six texture features (first-order uniformity, gray-level run-length matrix run percentage, neighborhood gray tone difference matrix coarseness, first-order energy, gray-level co-occurrence matrix cluster prominence, and neighborhood gray tone difference matrix strength) the diagnostic performance (AUC/sensitivity/specificity/accuracy) was 0.85/77.9%/82.8%/80.3%. The diagnostic performance of the radiomic model was significantly higher compared to the consensus read of the T2-FLAIR mismatch sign (*p* = 0.04), with a 21% increase in sensitivity and a 13% increase in overall accuracy while maintaining a similar (83%) specificity (Figure 3).

### 3.4. External Validation Results

The diagnostic performance (AUC/sensitivity/specificity/accuracy) of the T2-FLAIR mismatch sign was 0.66/62.5%/70.0%/66.3%. Using the T2-FLAIR mismatch sign, a total of 10/18 (55%) of 1p/19q co-deletion statuses were correctly identified.

The diagnostic performance (AUC/sensitivity/specificity/accuracy) of our radiomic model was 0.80/87.5%/89.9%/88.8%. Using the radiomic model, a total of 16/18 (89%) of 1p/19q co-deletion statuses were correctly identified.

### 3.5. Neuroradiologist + Radiomics

The initial diagnostic performance without the use of radiomic assistance was 0.67/53%/80%/67% (AUC/sensitivity/specificity/accuracy). Following the use of the radiomic model as a computer-assisted tool, the diagnostic performance improved to 0.74/65%/86%/75% (AUC/sensitivity/specificity/accuracy). Specifically, using the radiomic-assisted tool resulted in an increase in the correct identification of 13 cases in comparison to the initial interpretation with 11/13 correctly changed from matched to mismatch and 2/13 correctly changed from mismatch to match, which translates to approximately 12% increase in sensitivity and 2% increase in specificity. There were 11 cases of missed opportunities, where all cases were initially scored as matched and remained so despite the correct prediction of mismatch by radiomics.

ROC comparison showed significant improvement after using the radiomic-assisted tool (*p* = 0.019) (Figure 3).

Figure 4 shows an example of two instances where the initial incorrect assignment of the 1p-19q status was correctly changed after using a predictive texture model such as CAT.

## 4. Discussion

Results from the current study indicated that the proposed radiomic model outperformed the diagnostic accuracy of the T2-FLAIR mismatch sign in the prediction of 1p/19q co-deletion status and the diagnostic performance of the proposed radiomic model appears concordant with recently reported investigations [15,20]. In a study by Han et al. [20], radiomic-based prediction showed excellent performance in the detection of co-deletion status (AUCs of 0.89). They also showed that the integration of clinical factors such as demographic data and imaging features such as border cleanliness, homogeneity, and the presence of cysts did not improve the diagnostic accuracy over the radiomic-only model. Zhou et al. [15] combined radiomic features with Visually Accessible Rembrandt Images (VASARI) features (eloquent brain involved, proportion of enhancing tumor, proportion of non-enhancing tumor, cysts, multifocal or multicentric, definition of the non-enhancing margin, proportion of edema, diffusion, enhancing tumor crosses midline, and satellites) and only minimally improved the predictive performance of AUC 0.88 to 0.89 over the radiomics-only model. Additionally, to the best of our knowledge, the current study is one of the first to report comparative diagnostic performance of a radiomic-based model prediction to the T2-FLAIR mismatch signature in the same cohort of patients. The current results suggest that the added value obtained through the use of a radiomic model comes through the improved sensitivity without sacrificing the specificity known to accompany simple features as in T2-FLAIR mismatch. Specifically, data suggests the sensitivity for the prediction of 1p/19q co-deletion status was improved by approximately 21% in the training dataset and 25% in the testing data sets with the use of radiomics when compared to the use of T2-FLAIR mismatch alone. Importantly, results from the current study suggests that the use of radiomics as a computer-assisted tool can improve the diagnostic performance of neuroradiologists when estimating the 1p/19q status, with approximately a 12% increase in sensitivity, and even a 2% increase in specificity, when compared to the use of T2-FLAIR mismatch alone (*p* = 0.019).

While the use of T2-FLAIR mismatch has emerged as a promising and simple tool for the prediction of 1p/19q co-deletion status, its broad clinical utility has been impeded by a relatively low sensitivity and modest interobserver agreement. The sensitivity of the T2-FLAIR mismatch sign for the prediction of 1p/19q co-deletion status in the current study (57%) was comparable to prior reports, ranging from 37 to 55% [8,11]. These results suggest that roughly only half of the patients with 1p/19q non-co-deletion appear to exhibit the T2-FLAIR mismatch sign. The low sensitivity of T2-FLAIR mismatch may be due to prevalence of heterogeneity within and across gliomas, since a large majority may not demonstrate a homogenous suppression of fluid signal on FLAIR images, and hence lower sensitivity in using T2-FLAIR mismatch as defined by current convention [8,9]. Thus, the ~20% improvement in sensitivity with the current radiomic model and 12% increase in sensitivity when using the radiomic-assisted model may be a valuable addition to the current use of T2-FLAIR mismatch in routine clinical practice. In addition to increased sensitivity, the current study confirmed the relatively low interobserver agreement for the use of T2-FLAIR mismatch to identify 1p/19q status (k = 0.49). While the current study reported slightly lower agreement compared with prior reports, which range from 0.68 to 0.89 [8,10,11,28], the observed variability and subjective determination of T2-FLAIR mismatch further supports the need for a reliable and quantitative tool for quickly predicting 1p/19q status as suggested by the current radiomic approach. Lastly, it has been reported that FISH testing for 1p/19q can occasionally be erroneous which is an inherent limitation as the baseline 1p/19q status in this study was determined through FISH testing [22].

As more radiomic analysis is being incorporated in clinical applications, radiomic prediction of 1p/19q co-deletion status may be beneficial in guiding treatment management. With models with increased sensitivity, less LGGs with the 1p/19q co-deletion status may need to undergo invasive brain biopsy. Further, radiogenomics can assist in the optimization of patient management with regard to counseling and tumor classification when histologic characterization is not possible. In this study, we assessed several popular machine-learning approaches such as logistic regression, support vector machine, Adaboost, and Random Forest during the model development stage based on similar strategies in our previous studies [29,30] and other machine-learning radiomics studies [31,32]. We picked XGBoost as our final classifier based on its high performance, better automated feature selection, and robustness. Further investigation into classifiers is warranted as new algorithmic approaches are evaluated.

It is important to note that the current study was relatively small and retrospective in nature, only involving two institutions. A similar approach with a larger sample size, drawn from a pool of multicenter data, would improve statistical power and the generalizability of the current findings. Moreover, with a larger cohort, a deep-learning approach may be developed to provide an end-to-end solution. Importantly, the current study adheres to the standardized brain tumor imaging protocol [23], which should allow this approach to be generalized for use in clinical trials and for clinical management in centers that adhere to this international standard. It is also important to note that expression of T2/FLAIR mismatch is speculative and subjective, and further understanding of individual texture features correlating with 1p/19q co-deletion in relation to the mismatch will require further investigation.

## 5. Conclusions

The current study demonstrates the added value of a radiomic model to increase the sensitivity and diagnostic performance of determining 1p/19q status in IDH mutant gliomas when compared to the highly specific T2-FLAIR mismatch signature.

## Figures and Tables

**Figure 1 cancers-15-01037-f001:**
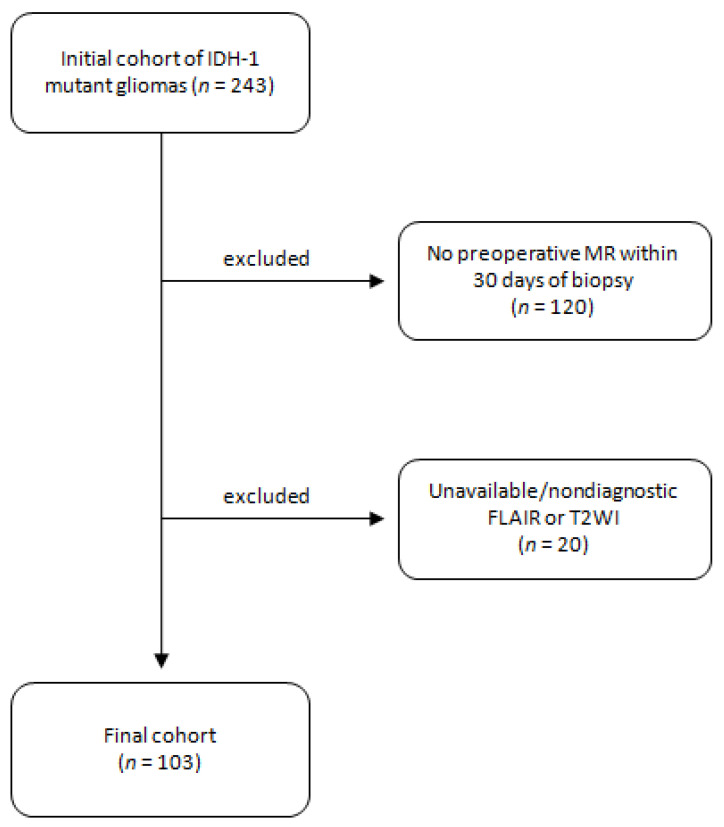
Study diagram with initial cohort and inclusion/exclusion criteria.

**Figure 2 cancers-15-01037-f002:**
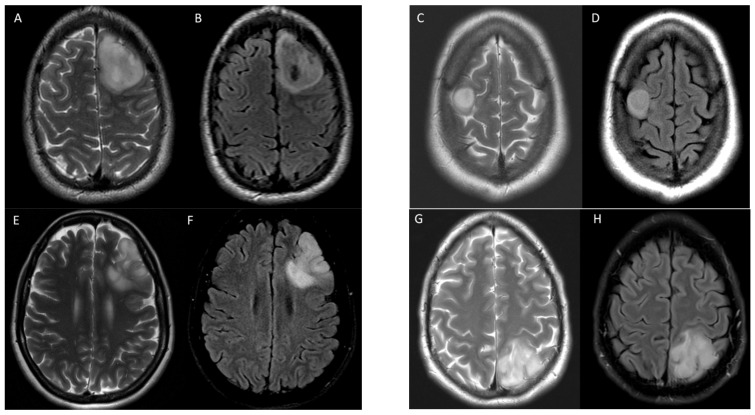
Examples of T2-FLAIR match and mismatch in 4 patients with IDH1 mutant gliomas with various 1p/19q co-deletion status. Axial T2 (**A**) and FLAIR (**B**) images demonstrate T2-FLAIR mismatch in a 25-year-old male with 1p/19q non-co-deletion correctly identified in a true positive instance. Axial T2 (**C**) and FLAIR (**D**) images demonstrate T2-FLAIR mismatch in a 58-year-old female with 1p/19q co-deletion in a false positive instance. Axial T2 (**E**) and FLAIR (**F**) images show T2-FLAIR match in a 55-year-old male with 1p/19q non-co-deletion in a false negative instance. Axial T2 (**G**) and FLAIR (**H**) images show T2-FLAIR match in a 60-year-old female with 1p/19q non-co-deletion correctly identified in a true negative instance. All cases were correctly identified by radiomic model.

**Figure 3 cancers-15-01037-f003:**
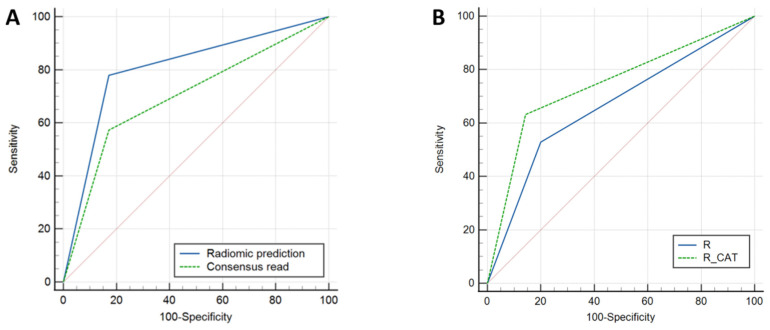
Receiver-operating characteristic (ROC) curves for prediction of 1p/19q co-deletion status. (**A**) Comparison of radiomics model to consensus T2-FLAIR mismatch with AUCs of 0.80 and 0.70, respectively (*p* = 0.044). (**B**) Comparison of radiologist T2-FLAIR mismatch alone to combined radiologist + radiomic (computer-assisted tool) with AUCs of 0.66 and 0.74, respectively (*p* = 0.019).

**Figure 4 cancers-15-01037-f004:**
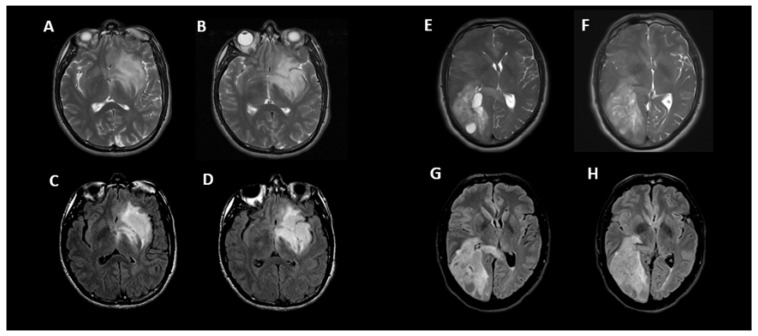
Two examples of correct changes made to T2-FLAIR mismatch evaluation with CAT. Axial T2 (**A**,**B**) and FLAIR (**C**,**D**) images in a 33-year-old male with IDH1 mutant glioma and non-co-deleted 1p/19q. During the initial interpretation, this was read as T2-FLAIR match but during the 2nd interpretation in conjunction with radiomic predicted result, this was correctly changed to T2-FLAIR mismatch. In this case, the heterogeneity of signal in different parts of tumor may have contributed to the initial error. Although there is relative matched signal between (**A**,**C**), there is subtle suppression of signal within the lower part of this tumor seen on (**B**,**D**) images. Axial T2 (**E**,**F**) and FLAIR (**G**,**H**) images in a 50-year-old male with IDH1 mutant glioma and co-deleted 1p/19q. During the initial interpretation this was read as T2-FLAIR mismatch; however, in 2nd interpretation in conjunction with radiomic predicted result, this was correctly changed to match. In this case, the prevalence of multiple small cystic components and their suppression on FLAIR images may have contributed to the initial error; however, when comparing (**F**–**H**), it is apparent that the majority of signal is matched between T2 and FLAIR.

**Table 1 cancers-15-01037-t001:** Demographic information for study cohort.

	1p19q Co-Deleted (*n* = 35)	1p19q Non-Co-Deleted (*n* = 68)	*p* Value
Age (mean/SD)	43/13	40/13	0.91
Sex (M/F)	12/23	23/45	0.93
Location (F/P/T/O/C) *	23/7/4/0/1	28/11/18/0/4	0.81
T2-FLAIR mismatch (Y/N)	6/29	39/29	<0.05

* F: frontal lobe, P: parietal lobe, T: temporal lobe, O: occipital lobe, C: cerebellum.

## Data Availability

The data presented in this study are available in this article.

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
