# Peer review of "Multi-Parametric Radiomic Model to Predict 1p/19q Co-Deletion in Patients with IDH-1 Mutant Glioma: Added Value to the T2-FLAIR Mismatch Sign"

_cancers, 2023, doi:10.3390/cancers15041037_

Round 1

Reviewer 1 Report

  1. There is a growing need for non-invasive imaging methods to identify genetic changes in the brain, particularly after the World Health Organization updated its classification system for central nervous system tumors in 2021. 1p/19q codeletion has been identified as a key prognostic predictor and is now used to diagnose oligodendrogliomas. Another imaging marker called T2/FLAIR mismatch has been shown to be a highly specific but relatively insensitive method for detecting IDH-mutated astrocytomas with 1p/19q non-codeletion. However, it is not clear whether T2/FLAIR mismatch is effective at detecting 1p/19q codeletion. The authors of this study suggest that their radiomic model may be able to improve the sensitivity of T2/FLAIR mismatch in detecting 1p/19q codeletion. This would need clarification to avoid confusion among the readers.
  2. Need more details regarding pre-processing and normalization methodology, so this study can be replicated and tested by other research groups.
  3. Overall, the methodology for conducting this study is well documented.  

Author Response

  • There is a growing need for non-invasive imaging methods to identify genetic changes in the brain, particularly after the World Health Organization updated its classification system for central nervous system tumors in 2021. 1p/19q codeletion has been identified as a key prognostic predictor and is now used to diagnose oligodendrogliomas. Another imaging marker called T2/FLAIR mismatch has been shown to be a highly specific but relatively insensitive method for detecting IDH-mutated astrocytomas with 1p/19q non-codeletion. However, it is not clear whether T2/FLAIR mismatch is effective at detecting 1p/19q codeletion. The authors of this study suggest that their radiomic model may be able to improve the sensitivity of T2/FLAIR mismatch in detecting 1p/19q codeletion. This would need clarification to avoid confusion among the readers.

Thank you for this comment. We have edited the discussion to clarify that the radiomic model and the radiomic-assisted model both improved the sensitivity of predicting 1p/19q codeletion status when compared to and used in conjunction with the T2/FLAIR mismatch sign.

  • Need more details regarding pre-processing and normalization methodology, so this study can be replicated and tested by other research groups.

An automated and standardized normalization step is performed as part of preprocessing for each case using the OLEA software. We have edited the statement in the methods section for clarification.

  • Overall, the methodology for conducting this study is well documented.  

Thank you.

Reviewer 2 Report

The present study investigated on the prediction of 1p/19q codeletion through T2/FLAIR mismatch identification by means of neuroradiologists traditional analysis and machine learning (ML) method. Patients’ number is relatively low but comparable to previous researches, the presence of an external validation cohort is appropriate for this kind of study.

I have few comments:

-       The paper focus on the importance of T2/FLAIR mismatch for 1p/19q codeletion prediction, but we are not sure if the ML method based its decision on this feature since the output was the prediction of gene codeletion. A more straightforward method would consider the mismatch identification as output and then verify if this result is in line with tumor molecular profile. Although the combination of features considered for the analysis are mostly based on grey levels, at the moment the only clear result is the codeletion prediction with high accuracy and not the T2/FLAIR mismatch utility, especially if false positive and negative are shown in figures. I would ask the authors to consider redoing the analysis with different output.

-       Why the authors chose eXtremeGradientBoosting as a classifier? It is true that ensemble classifiers showed superior performances in medical field (see 10.3414/ME0543). A recent research showed good xGB performances in predicting overall survival (see 10.3389/fonc.2021.601425), while other (like Adaboost) were more fit for molecular status prediction (although 1p/19q was not evaluated). Did the authors consider these previous research for their classifier selection?

-       In a similar fashion, why did the authors did not consider a deep learning algorithm setting T2/FLAIR mismatch as an output? Neural networks proved to predict glioma mutations with high accuracy (see 10.1093/neuonc/noaa177, 10.3390/jpm11040290 and 10.1093/noajnl/vdaa066). Please further expand discussion considering this point.

-       Neuroradiologist + Radiomics section should be separated between methods and results. In methods please insert the description of the experiment, in the results section just the results.

-       Experience of the two neuroradiologists don’t match between methods (4 and 5 years) and results (6 and 8 years) sections. Please correct.

-       Figure 4 is difficult to read. Please consider arranging it horizontally by regrouping A, B, C, D on the left and (separated by a space) E, F, G, H on the right.

Round 2

Author Response

Thank you for these responses. All edits have been made as suggested on the manuscript. Please see separate updated manuscript attachment.